# Experimental phage evolution results in expanded host ranges against antibiotic resistant *Klebsiella pneumoniae* isolates

Pooja Ghatbale[1], Alisha Blanc[1], Andrew Sue[1], Jesse Leonard[1], Monica Bates[1], Andrew G. Garcia[1], Joshua Hensley[2,3], Danielle Devequi Gomes Nunes[4], Nicole Hitchcock[1], Job Shiach [5], Roberto Bardaró[4], Govind Sah[1], Chandrabali Ghose[6], Katrine L. Whiteson [7], Robert T. Schooley [5], Richard Allen White III[2,3], Ana G. Cobián Güemes[1], Justin R. Meyer [8] & David T. Pride[1,5] ✉

Resistance to antibiotics is approaching crisis levels for organisms such as the ESKAPEE pathogens (includes Enterococcus faecium, Staphylococcus aureus, Klebsiella pneumoniae, Acinetobacter baumannii, Pseudomonas aeruginosa, Enterobacter spp., and Escherichia coli) that often are acquired in hospitals. These organisms sometimes have acquired plasmids that confer resistance to most if not all beta-lactam antibiotics. We have been developing alternative means for dealing with antibiotic resistant microbes that cause infections in humans by developing viruses (bacteriophages) that attack and kill them. One of these pathogens, K. pneumoniae, has one of the highest propensities for antimicrobial resistance. We identified many phages that have lytic capacity against limited numbers of clinical isolates, and through experimental evolution over the course of 30 days, were able to vastly expand the host ranges of these phages to kill a broader range of clinical K. pneumoniae isolates including MDR (multi-drug resistant) and XDR (extensively-drug resistant) isolates. Most interestingly, they were capable of inhibiting growth of clinical isolates both on solid and in liquid medium over extended periods. That we were able to extend the host ranges of multiple naïve antibiotic resistant *K. pneumoniae* through experimental phage evolution suggests that such a technique may be applicable to other antibiotic-resistant organisms to help stem the tide of antibiotic resistance and offer further options for medical treatments.

Antibiotic resistant pathogens have been on the rise around the globe in part due to the overuse of antibiotics and the adaptation of bacteria to the constant exposure to those antibiotics[1-3]. Because many available antibiotics work through similar mechanisms of action, resistance to one antibiotic often results in the development of resistance to multiple different antibiotics within that antibiotic class. That is why the widespread use of antibiotics in agriculture and in animal husbandry may be so profound, as even if they are not the same antibiotics as are used in humans, the fact that they often represent the same classes of antibiotics that are used in humans can result in cross-resistance in the pathogens that are found in humans to the antibiotics that cause disease in humans[4].

---

To combat trends in antibiotic resistance, there are pipelines to develop antibiotics based on novel mechanisms[5], but also searches for novel antimicrobials. Another means for addressing the issue is the pursuit of a solution that is now over 100 years old but was largely abandoned in Western cultures due to its replacement by antibiotics[6,7]; that solution is bacteriophages. Phages are viruses that specifically target and kill bacteria and generally pose no threat to the human host[8]. They were replaced by antibiotics because they generally lack the range of killing activity of antibiotics[9], are more difficult to work with[10], and knowledge of their pharmacodynamics and pharmacokinetics is generally lacking[11]. Phage technology has recently seen a resurgence in Western medicine[12] given the surge in antibiotic resistant microbes and the relative lack of available alternatives for physicians. Ongoing clinical trials evaluating the safety of phage therapy are an important step towards enabling its broader clinical application. Also, given recent advances in technology, scientists have been poised to make advances in phage technology that were not available when antibiotics were largely replacing them as the treatments of choice for bacterial infections.

Several infections are commonly associated with patients in the hospital and sometimes even in the community, and many of them are caused by ESKAPEE pathogens (including *Enterococcus faecium*, *Staphylococcus aureus*, *Klebsiella pneumoniae*, *Acinetobacter baumannii*, *Pseudomonas aeruginosa*, *Enterobacter spp.*, and *Escherichia coli*)[13]. These pathogens have a high propensity for antimicrobial resistance and are found across the globe[14]. Many of them have recently been the targets of phage related treatments[15].

In this work, we focused on the pathogen *Klebsiella pneumoniae* and adapted existing co-evolutionary techniques to evolve phages that had implemented strategies to overcome the development of rapid *K. pneumoniae* resistance in clinical strains. In particular, we were seeking to evolve Klebsiella phages that had expanded and prolonged activity against MDR (multi-drug resistant) and XDR (extensively-drug resistant) clinical isolates of *K. pneumoniae*.

## Results

### Isolation of bacteriophages

We developed a collection of bacteriophages that infect and lyse both *Klebsiella pneumoniae* and *Klebsiella* spp. isolates with the goal of determining whether these isolates would have activity against MDR and XDR clinical isolates. These phages were identified in the continental USA (Table S1) using our standard phage isolation techniques[16] from various environmental sources. We identified and sequenced a total of 11 unique phages belonging to 4 different families (Table S1).

Their genomes ranged in size from 37.9 Kbp to 177.7 Kbp. Each of these phages was closely related to previously characterized phages, having high levels of identity to phage sequences stored in the NCBI database (Table S2). The phages were found to represent four distinct phylogenetic clades, with Beam, KL35, Rec, QTY, and Turmeric making up one clade, APV, Ace, and LK1 making up another, and LK2 and Chai representing the third (Fig. S1). Phage LK3 was an outlier, as it was not genetically similar to any of the other phages included in this study (Fig. S1).

We characterized the morphologies of the phages via transmission electron microscopy (TEM) and identified a range of different morphologies (Fig. 1). For example, phages Ace, APV, Beam, QTY, Rec, Turmeric, LK1, and KL35 all had morphologies consistent with those of myoviruses (Panels A–F, H, K), while LK2 had a morphology consistent with podoviruses (Panel I), and others were consistent with siphoviruses (Panels G, J). These data confirm that we identified several different morphologies amongst the 11 different viruses we characterized.

### Examination of host ranges

We evaluated the host ranges of the 11 phages against a group of 59 clinical isolates of *K. pneumoniae* and *K.* spp. to determine whether they were capable of lysing them. Of the 59 isolates tested, 20 of them were relatively susceptible to antibiotics, while 39 of them were known to possess Extended Spectrum Beta Lactamases (ESBLs) or other antibiotic resistance genes (Table S3). Of the 39 ESBL *Klebsiella* isolates, 4 were Carbapenem Resistant Enterobacteria (CRE), and of the other 20 Klebsiella isolates, 1 was a CRE. These isolates were derived from blood, urine, respiratory tracts, or other body sites from human subjects. Some of the 11 phages had little if any activity against the Klebsiella isolates, including phages LK3, LK2, and Chai, while others, including KL35, Tumeric, Rec, QTY, APV, Ace, and Beam had variable activity against the isolates (Fig. S2).

### Experimental evolution

Building on a previous study that demonstrated the effectiveness of coevolutionary training for laboratory adapted *E. coli* isolates[17], we hypothesized and tested that coevolutionary training might have an additional effect: broadening phage host ranges for clinical *K. pneumoniae* isolates alongside improving their effectiveness. We predicted that as phages engaged in an arms race with their hosts, they would either accumulate counter-defenses usable against multiple bacterial strains or evolve reduced specificity to overcome resistant hosts. The primary differences between the two protocols were the use of clinical isolates compared to laboratory-adapted isolates, the fact that

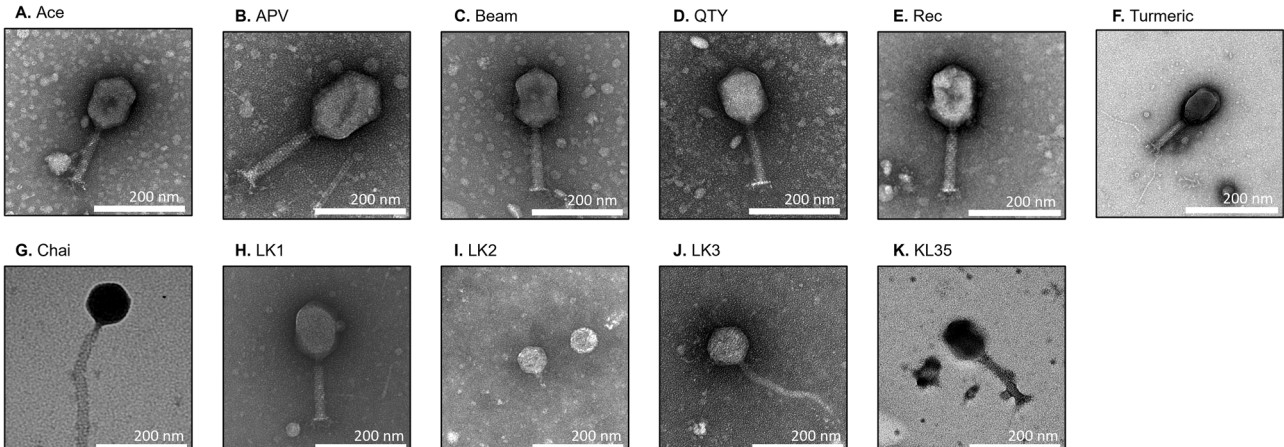

**Fig. 1 | Morphology of Klebsiella phages examined via Transmission Electron Microscopy (TEM).** TEM images of negatively stained phages with 2% of uranyl acetate (A–K). Scale bars are 200 nm represented on each panel.

**A.** Schematic representation of coevolution experiments

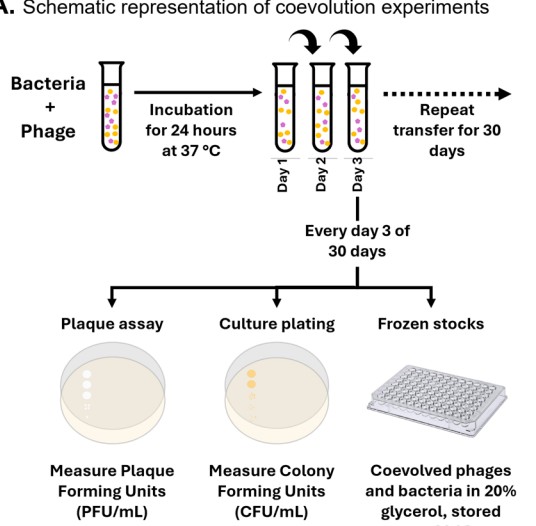

**B.** Schematic representation of testing evolved phages

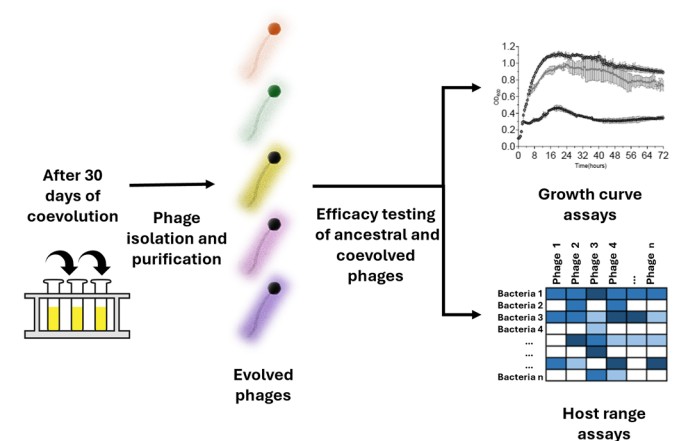

Fig. 2 | **Overview of experimental design. A** Phages were coevolved with various clinical strains of *K. pneumoniae* with an MOI of 10 at 37 °C for 30 consecutive days and each day 1% of previous day phage and bacterial coevolved population was transferred to fresh media. The population dynamics of evolved phages and bacterial hosts were estimated based on plaque forming units (PFU) and colony forming units (CFU), respectively. Each phage and host was isolated and saved on every day 3 of coculturing. **B** After 30 days of coevolution study, the corresponding evolved phages were tested for kinetic and lytic efficiency by performing liquid growth curve and host range assays.

Klebsiella isolates had mucoid phenotypes compared to non-mucoid *E. coli*, and the evolution process between host and phage took place for a full 30 days before the phages were recovered (Fig. 2A). Specimens were transferred into fresh media on a daily basis to prevent nutrient depletion, and specimens were titered every 3 days to evaluate whether the phages-maintained viability (Fig. S3). We then isolated the phages on day 30 and evaluated whether they had different host ranges than those that we started with and if they presented higher growth inhibition than the ancestral phages (Fig. 2B).

### Host range expansion

We focused on two of our phages with relatively modest host ranges (Fig. S2), phages Ace and APV (belonging to the family Straboviridae) for our phage evolution/host range expansion experiments. We selected phages with modest host ranges to give them the opportunity for improvement. Prior to the expansion experiments, phage APV had some lytic activity against 27.12% of the isolates tested, while phage Ace had lytic activity against 42.37% of the tested isolates. After 30 days of the experimental evolution, in the spot titer tests, we found substantial differences in the lytic capacity of phage APV before and after the experiment. For example, different trained phage APV isolates varied in their lytic capacity, from having some lytic capacity against 16.95% of the isolates to 61.02% of the isolates. Overall, the host range of the co-evolved phages increased in four experiments and decreased in two experiments (Fig. 3). We found similar results for phage Ace, where the post-evolution phage isolates ranged in activity from 30.51% to 59.32% of the *K. pneumoniae* isolates, showing an expansion of host range in three experiments, a decrease in host range in two experiments, and remained the same in one experiment. There was host-range expansion identified against both ESBL isolates and non-ESBL isolates. These data strongly suggest that the 30-day process of phage-host evolution resulted in a set of phages with improved capabilities in lysing their *K. pneumoniae* hosts. Host range expansion was observed to *K. pneumoniae* clades for which the ancestral phages showed no lysis. Coevolved APV phages expanded their host ranges to four clades that were not lysed by the ancestral phage. Coevolved Ace phages expanded their host range to four clades that were not lysed by the ancestral phage (Fig S4). This host range expansion also included lysis of putative O-antigen and capsule serotypes not previously infected.

### Evaluation of growth dynamics

Phage training involves co-incubating bacterial host and phage together over successive generations so that the phage adjusts to changes in the host to allow it to survive common host adaptations[18]. Such training experiments have previously been performed for *E. coli*, *K. pneumoniae*, *P. aeruginosa*, and *L. monocytogenes*[19–22]. While examination of phage activity using spot assays can provide an understanding of lytic potential, we believe the ability to inhibit growth longitudinally in broth media is a better approximation of the ability of phages to kill their hosts while overcoming their ability to develop resistance rapidly. We examined the ability of the evolved phages Ace and APV compared to the ancestral phages Ace and APV to determine whether the evolutionary process resulted in better adapted phages for longitudinal inhibition of *K. pneumoniae* host growth. We found that over the 72 h, the trained phages were superior to the ancestral phages in suppressing the growth of the *K. pneumoniae* isolates (Fig. 4). Indeed, in 10/12 examples, the trained phages demonstrated better suppression of *K. pneumoniae* growth longitudinally. We examined whether the suppression was statistically significant via area under the curve analysis (AUC) and found that in 7 of the 12 examples (58%), the longitudinal suppression of the *K. pneumoniae* isolates was significant (Fig. 5). These isolates were all highly susceptible to antibiotics and generally would not be the target of treatment with phages. To decipher whether the evolved versions of phages Ace and APV may have utility against antibiotic-resistant *K. pneumoniae* isolates, we would need to decipher whether those phages would be capable of suppressing MDR and XDR *K. pneumoniae* isolates.

### Inhibition of MDR and XDR isolates

To determine whether evolved phages Ace and APV had greater activity against MDR and XDR *K. pneumoniae* isolates, we tested them against our collection of MDR and XDR isolates in broth media over 72 h. We found that in all cases, the trained phages were superior to their untrained counterparts in their ability to suppress growth of their hosts (Fig. 5). This trend of suppressing the growth of their hosts extends when evaluating a larger group of clinical *K. pneumoniae* isolates and even to CRE isolates (Fig. S6). We also evaluated whether the suppression of the *K. pneumoniae* hosts with these trained phages was statistically significant over the 72 h using AUC analysis and found that

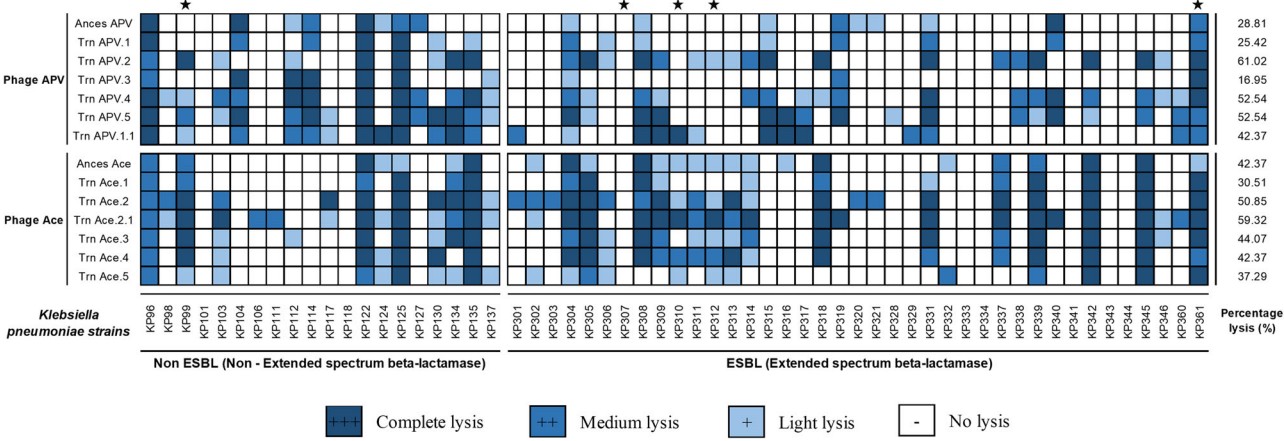

**Fig. 3 | Host range and total percentage killing of the phages against the collection of 59 clinical strains of *K. pneumoniae*.** Phage lysate (5 μl) with $10^7$ titer was spotted on a bacterial lawn. After overnight incubation, the plates were examined for lysis and no lysis. The dark blue boxes indicate complete lysis, medium to light blue boxes indicate medium to light lysis and boxes with no color represent no lysis. The isolates were grouped based on extended-spectrum beta-lactamases (ESBLs). CRE (Carbapenem Resistant Enterobacteriaceae) isolates are indicated with a star.

21/28 (75%) of the isolates were indeed significantly suppressed (Figs. S7, S8). These data indicate that the process of evolving Klebsiella phages with their clinical hosts results in some phages with significantly improved suppression profiles of MDR and XDR isolates over at least 72 h.

## Phage genetic changes

We examined the phages that were coevolved over the 30-day experiment to identify what differences may have occurred that led to the improved suppression profiles and the extended host ranges that led to the suppression of the MDR and XDR Klebsiella isolates. All trained Ace phages were similar on the whole genome level by average nucleotide identity level with less than 0.01% difference post 30-day evolution (Panel A in Figs. 6 and S9). APV phages had greater overall genome differential change at 0.02%, except APV.1 and APV.5 (Panel B in Figs. 6 and S9). Phage ACE.3 had a 165 nucleotide (nt) deletion in an intergenic region. All trained phages had at least one nonsynonymous mutation in the L-shaped tail fiber protein, which is responsible for phage recognition and binding to host cells. Indeed, the majority of the mutations observed were concentrated in the tail fiber and baseplate regions of both phages (Figs. 6 and S10), each of which may be involved in host recognition and binding. These data suggest that the coevolution process resulted in changes in phage recognition/binding, which may be responsible for the phenotypic changes in host suppression we observed.

## Discussion

Antimicrobial resistance among bacteria has become a rising and persistent global threat[23]. Particularly amongst the ESKAPEE pathogens, there is a critical need to pursue alternative options outside of conventional antibiotics for the treatment of the life-threatening infections caused by these microbes. *Klebsiella pneumoniae* is prominent amongst these microbes, as it has a propensity to develop multidrug resistance through the acquisition of ESBL plasmids, and extensive drug resistance through the presence of KPCs (*Klebsiella pneumoniae* Carbapenemases)[24]. It also produces a highly mucoid outer coating that can render the organism difficult to access and more resilient to treatment[25]. Because bacteriophages often are limited in their host ranges to select bacterial isolates[26], and resistance to them often develops rapidly[27], they often are at a distinct disadvantage compared to conventional antibiotics when considering treatment for bacterial pathogens. However, given that many bacterial pathogens such as *K. pneumoniae* are becoming resistant to most conventional antibiotic therapies, phage therapies are now being revisited.

Similar to a prior study by our groups[17,28] that demonstrated that we could significantly extend the host ranges of laboratory adapted *E. coli* isolates using experimental evolution techniques, we extended similar techniques to a group of MDR and XDR clinical Klebsiella isolates. We found that the same principles that apply to microbes that have spent years adapting to laboratory conditions also apply to microbes that are recently derived from patients with clinical infections. We evolved a small group of phages against clinical *K. pneumoniae* isolates and found that the trained phages significantly improve in their ability to inhibit clinical MDR and XDR *K. pneumoniae* isolates longitudinally (Figs. 4, 5). This change coincides with an expansion in host ranges for the phages (Fig. 3), a presumed reduction in the ability of the host to develop rapid resistance, and significant changes in the phages that allowed them to infect their host over extended time periods. Most of the changes that were observed were in a gene cluster of five tail fiber related proteins and a holin (Fig. 6). In both training experiments, the protein that accumulated the most changes was the tail fiber adhesin. This suggests that the observed changes played a role in phage adhesion to bacteria.

Because *K. pneumoniae* represents such a significant pathogen in communities and hospitals, and of its tremendous capacity for both MDR and XDR resistance to antibiotics, it represents an ideal pathogen to target with alternative means such as phages for medical treatments. We previously developed a targeted method for evolving laboratory adapted *E. coli* for host range expansion; however, the methods detailed here provide a template for targeting MDR and XDR clinical pathogens with phages in a compressed time frame. Prior studies have examined training phenomena in other pathogens such as *P. aeruginosa* and *L. monocytogenes*[19,20] with some success, so other pathogens are definitely subject to the types of phage training protocols used in this study. While we developed this protocol to take place in as little as 30 days, it likely is possible to develop such phages with broad host ranges toward antibiotic resistant human pathogens in much shorter time frames. While this study only detailed this protocol for a single clinical ESKAPEE pathogen, we believe it provides a template for moving forward with targeting most all antibiotic-resistant pathogens for which there are available phages.

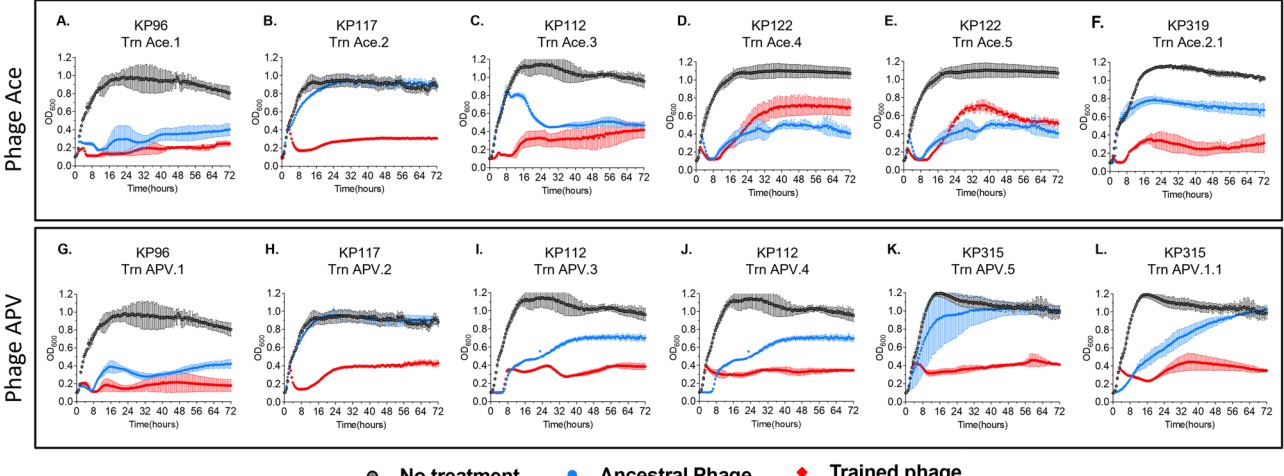

**Fig. 4 | Growth dynamics of coevolved hosts in presence of trained vs untrained phages at MOI of 1.** The phage and bacteria were coevolved together for 30 consecutive days, the trained phages after 30 days were isolated and determined their kinetic efficiency on their corresponding coevolved hosts by monitoring bacterial growth for 72 h at 37 °C. Growth curves are shown as the average of 3 separate biological replicates for each microbe and/or microbe/phage pairing with standard deviation bars in each panel (**A**–**L**).

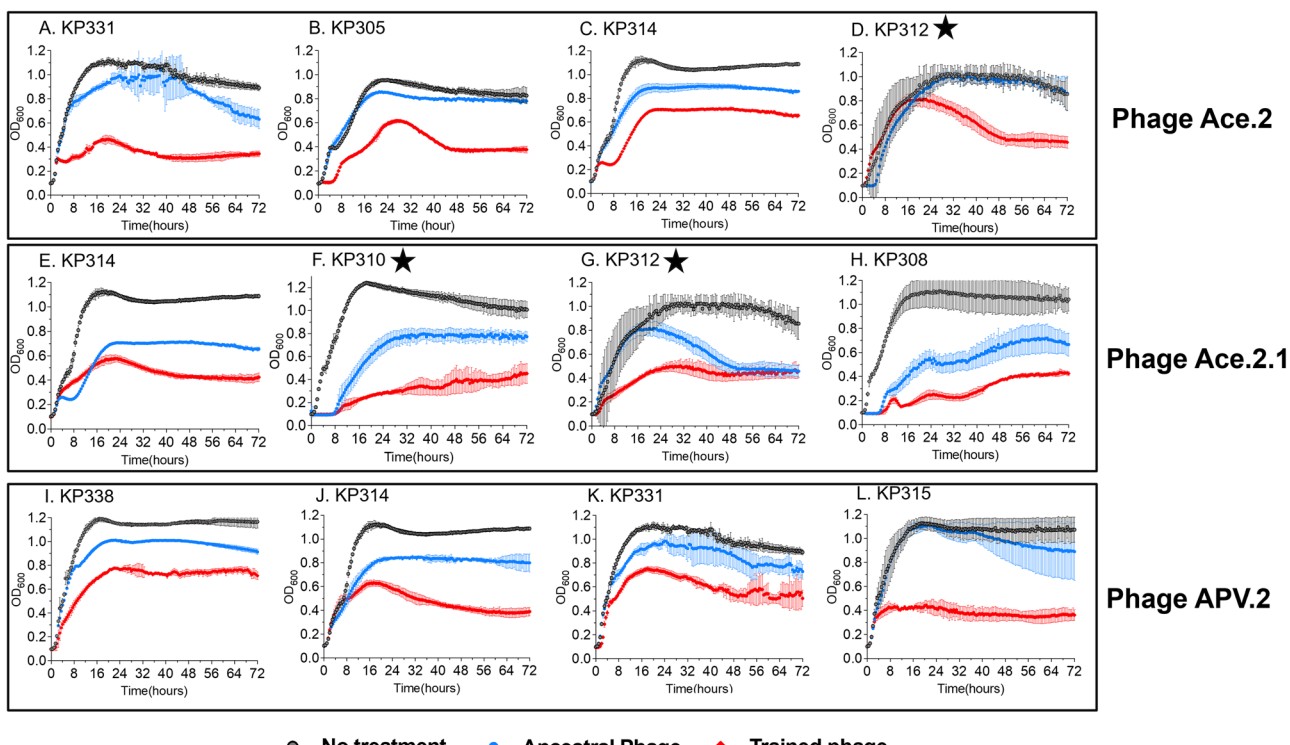

**Fig. 5 | Growth dynamics of ESBLs with most divergent trained phages.** Experiments were performed at MOI of 1, and phages trained in KP117 were used for all experiments (APV.2 and Ace.2). Bacterial growth was monitored for 72 h by measuring the OD600 every 15 min in a microplate reader. Experiments were performed at 37 °C. Note: Growth curves for CRE isolates are indicated with a star. Growth curves are shown as the average of 3 separate biological replicates for each microbe and/or microbe/phage pairing with standard deviation bars in each panel (**A**–**L**).

## Methods

### Strains and bacteriophages

The *Klebsiella pneumoniae* strains used in the study were isolated from patients at UC San Diego Health (Table S1). Strains were screened for antibiotic susceptibility using MicroScan Neg MIC 46 and MicroScan RUO panels[29]. The phages used in the study were previously isolated from water samples from various sources using the multiple enrichment protocol[30] (Table S1). For the training experiments, we used phages Ace and APV. All strains of *Klebsiella spp.* and phages were grown in Luria Bertani media (LB) at 37 °C. Phages were purified on host KP125 using a standard double agar overlay plaque assay[31].

### Electron microscopy observation of phage particles

To observe morphology of the phage particles, carbon coated grid (PELCO SynapTekTM Grids, product# 01754-F) was placed on a drop of

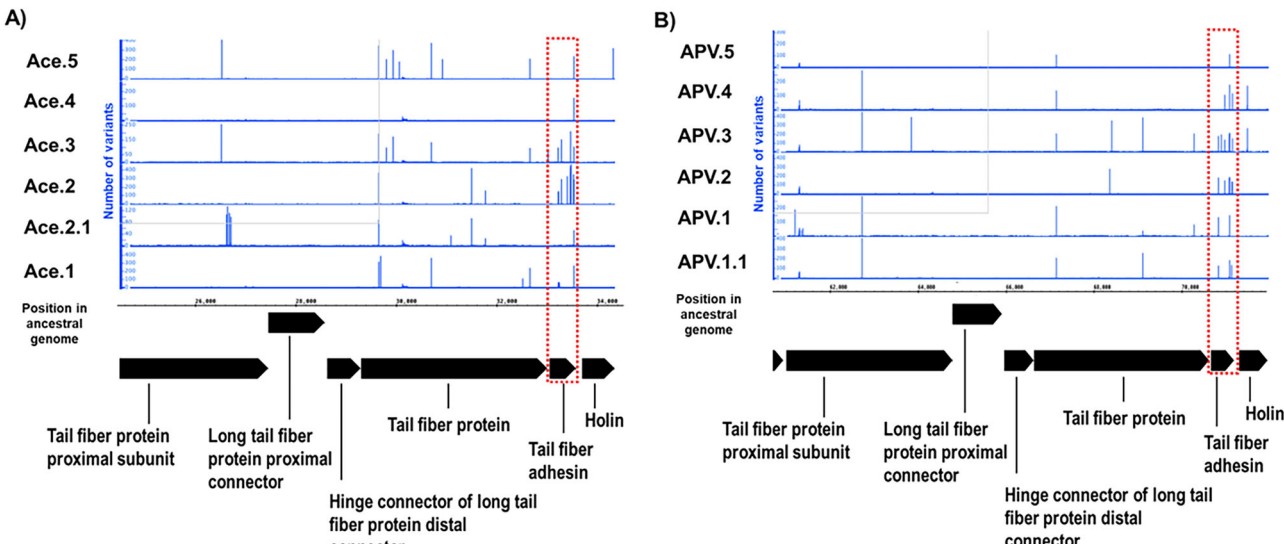

**Fig. 6 | Gene cluster with most variants in trained phages.** Panel (**A**) represents phage Ace and Panel (**B**) represents phage APV. Ancestral phages were aligned with trained phage genomes and visualized using the Integrated Genome Browser. The Integrated Genome Browser was then used to generate mismatch graphs from the BAM files showing the number of mismatched nucleotides compared to the reference sequence.

10 µL of freshly made high titer phage stock and the negative staining with 2% uranyl acetate (pH 4.0) was performed. Later, the phages were imaged by transmission electron microscopy (TEM) using Joel 1400 plus at University of California, San Diego–Cellular and Molecular Medicine Electron Microscopy Core.

### Phages whole genome sequencing
Total genomic DNA from the isolated bacteriophages using QIAamp UltraSens Virus kit (Qiagen catalog number 53706). Whole genome sequencing was performed using the iSeq100 platform (Illumina), producing 150 bp paired-end read format (2 × 150 bp).

### Phages genome assembly and annotation
Illumina reads were quality controlled, trimmed, and adapter sequences were removed using CLC Genomics Workbench (CLC Genomics, Qiagen, version 9.5.3). The reads were assembled into contigs using de-novo assembly algorithm on CLC Genomics Workbench. CheckV (Table S4) was used for quality control assessment of the phage genomes[32]. Each phage genome was annotated using MetaCerberus (v1.4)[33] using all databases option with Pyrodigal-gv[34,35].

### Bacteria whole genome sequencing
Genomic DNA was extracted from the clinical strains using DNeasy Blood & Tissue Kit (Qiagen, catalog number 69506). Whole genome sequencing was performed using the NovaSeq platform (Illumina), producing 150 bp paired-end read format (2 × 150 bp).

### Bacteria genome analysis
Genomes were de novo assembled using SPades, annotations were performed using BV-BRC bacteria annotation recipe. Phylogenies were constructed using BV-BRC Bacterial Genome Tree tool which utilizes a codon tree method that selects single-copy BV-BRC PGFams and analyzes aligned proteins and coding DNA from single-copy genes using the program RAxML. Parameters were set to 1000 genes with 0 deletions or duplications. The phylogenetic clades and annotations were performed using iTOL. Isolates serotype classification was performed using the in silico program Kaptive, which takes genome assemblies and compares the K and O loci to a curated reference database to find the best matching locus. A scoring matrix is used that considers sequence identity, coverage, missing genes, and extra genes to determine the confidence level of the match between the query genome and the reference locus.

### Phage-bacteria coevolution experiment
Phage coevolution experiments were conducted as previously described[17]. Briefly, we co-cultivated each different lytic phage, phage Ace or phage APV, with each one of several clinical strains of *Klebsiella pneumoniae* (KP319, KP315, KP122, KP96, KP117, or KP112) in 4 mL of Tris LB (1 Liter of 1X LB supplemented with 50 ml of 1 M Tris-base (pH 7.4), 0.2 ml of 1 M CaCl2 and 10 ml of 1 M MgSo4) in three replicates for 30 consecutive days. All strains were susceptible to the phages at the beginning of the experiments. Every three days, 1% of the population of each community was transferred into 4 mL of fresh medium. One milliliter aliquots were made of each of these cultures to estimate the bacterial and phage densities, and remaining aliquots were preserved by freezing at −80 °C in 20% v/v glycerol. After 30 days of training, the phages were isolated and plaque purified three times, and the stocks were stored at 4 °C for later use.

### Host range evaluation and liquid growth curve assays
We tested whether the trained phages had broadened their host range/ lytic and kinetic efficiency in comparison to their ancestral phages by evaluating host range using spot tests and by performing liquid growth curve experiments, respectively. Host range assays on the collection of Klebsiella ancestral and trained phages were carried out as previously described[36]. Briefly, overnight grown bacterial cultures were used to make a bacterial lawn of each isolate 4 µL of each phage stock (-10^7 PFU/mL) were spotted, and the plates were left to dry at room temperature for 30 minutes, followed by incubation at 37 °C for 24 h. The next day, a zone of lysis and no lysis was determined. The phage and bacterial co-cultivation assays were performed at MOI of 1. The bacterial cells from exponential phase were diluted to an OD600 of 0.1 in fresh LB broth. Each well within a 96 well plate was inoculated with 20 µL of phage + 60 µL of bacteria (OD600 0.1) and remaining volume of LB media was added to a total volume of 200 µL. The OD600 was measured every 30 min at 37 °C for 72 h.

### Genomic comparisons of ancestral and coevolved phages
Phylogenetic trees of phages were produced using ViPTree[37], and amino acid-resolved genome alignments were made using DiGAlign[38].

Reads from the trained phages were mapped against their ancestral derived-phage (i.e., Ace or APV) using Bowtie2 (very-sensitive option, version 2.5.2)[39]. The resulting SAM files were converted to BAM and indexed using SAMtools version 1.20[40]. The variants were called using BV-BRC[41] variation analysis and the SNP caller Free Bayes[42]. Variants including mismatches were visualized using Integrated Genome Browser with annotation obtained from MetaCerberus (version 10.0.1[43]). Average nucleotide identities (ANI) were calculated using the EZBioCloud ANI calculator[44].

### Ethics statement

All bacterial isolates collected in this study were approved by the University of California San Diego Administrative Panel on Human Subjects in Medical Research. This Study was categorized as exempt, indicating that it does not qualify as human subjects research, which does not require informed consent on behalf of the study subjects. It was categorized as exempt because it involved the use of existing records and specimens that were recorded in such a manner that the subjects could not be identified directly or through identifiers linked to the subjects.

### Reporting summary

Further information on research design is available in the Nature Portfolio Reporting Summary linked to this article.

## Data availability

All genomes sequenced in this study have been deposited in the NCBI GenBank database under accession numbers PQ621130, PQ621129, PQ621128, PQ621122, PQ621123, PQ529758, PQ621133, PQ621124, PQ621131, PQ621132, PQ621121, and PRJNA1189177. They can be found using the link https://www.ncbi.nlm.nih.gov/nucleotide followed by the accession number.

## Code availability

The code used in this work is publicly available on GitHub (https://github.com/raw-lab/) and at https://doi.org/10.5281/zenodo.17296047.

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

## Acknowledgements

This study was funded by Emily's Entourage 30311868 to DTP and the Howard Hughes Medical Institute Emerging Pathogens Initiative grant 30207345 to DTP and JRM. RAW and JH are supported by the UNC Charlotte Department Bioinformatics and Genomics start-up package from the North Carolina Research Campus in Kannapolis, NC. JH was further supported by the University of North Carolina at Charlotte Office of Undergraduate Research (OUR). We thank UCSD Clinical Microbiology for their contributions to this work. We also want to acknowledge the Integrated Genome Browser team of the Lorraine lab, including Nowlan Freese and Paige Kulzer, for their assistance with the genome plots. This publication includes data generated at the UC San Diego IGM Genomics Center utilizing an Illumina NovaSeq 6000 that was purchased with funding from a National Institutes of Health SIG grant (#S10 OD026929).

## Author contributions

D.T.P. conceived and designed the project, facilitated the acquisition of the bacteria and phages, examined the data and wrote the manuscript. J.R.M. assisted with the project design and conception. A.G.C.G. oversaw the experiments, processed the data, and edited the manuscript. P.G. performed most of the experiments and processed the majority of the data. A.B. performed many of the experiments and processed data. A.S. processed many of the datasets and constructed figures. J.L. imaged most of the bacteriophages. M.B. processed many of the clinical isolates. A.G. oversaw the development of the bacterial and bacteriophage collections. J.H. performed bioinformatic analysis of mutations in bacteriophages. R.A.W. oversaw the analysis of bacteriophage mutations. D.D.G.N. assisted with development of the bacteriophage collection. R.B. oversaw the development of the bacteriophage collection. N.H. assisted with bacteriophage acquisition and host range establishment. J.S. assisted with bacteriophage host range establishment. G.S. identified and processed many of the bacteriophages. C.G. assisted with project design and conception. K.W. assisted with project design and conception. RS assisted with project design and conception.

## Competing interests

The authors declare no conflicts of interest. R.A.W. III is the CEO of RAW Molecular Systems (RAW), LLC, but no financial, IP, or others from RAW LLC were used or contributed to this study.

## Additional information

[1]Department of Pathology, University of California, San Diego, CA, USA. [2]North Carolina Research Center, Department of Bioinformatics and Genomics, The University of North Carolina at Charlotte, Kannapolis, NC, USA. [3]Computational Intelligence to Predict Health and Environmental Risks, Department of Bioinformatics and Genomics, The University of North Carolina at Charlotte, Charlotte, NC, USA. [4]SENAI Institute Innovation (ISI) in Health Advanced System, University Center SENAI/ CIMATEC, Salvador, Bahia, Brazil. [5]Department of Medicine, University of California, San Diego, CA, USA. [6]Bioharmony, Inc., New York, NY, USA. [7]Department of Biology, University of California, Irvine, CA, USA. [8]Department of Ecology, Behavior and Evolution, University of California, San Diego, CA, USA. ✉e-mail: dpride@health.ucsd.edu

