## [Transparent Peer Review file · Nature Communications]

Experimental Phage Evolution Results in Expanded Host Ranges Against Antibiotic Resistant *Klebsiella pneumoniae* Isolates

Corresponding Author: Dr David Pride

Version 1:

Reviewer comments:

Reviewer #1

(Remarks to the Author)

While the manuscript entitled “Experimental Phage Evolution Results in Expanded Host Ranges Against MDR and XDR *Klebsiella pneumoniae* Isolates” addresses the potentially important concept of expanding phage host range through evolutionary approaches to target clinically resistant bacteria, it unfortunately lacks sufficient novelty to warrant publication in its current form.

1. Except using clinical isolates, the experimental methodology employed for phage evolution substantially overlaps with established protocols reported in previous studies (e.g. doi: 10.1126/science.adi5536). Actually the hypothesis of broadening host range through phage co-evolution has been mentioned in previous studies, e.g. doi: 10.1126/science.adi5536 and doi: 10.1128/AEM.02138-20. It is not surprised to see that the evolved phages would expand host range.
2. The genetic mutations in the evolved phages were found related to the L-shaped tail fiber protein, affecting host recognition and binding mechanisms. The mechanism is also similar to that found in previous studies (e.g. doi: 10.1126/science.adi5536, doi: 10.1128/AEM.02138-20).
3. Based on previous studies, co-evolution of phage and bacteria will generate a mixture of different evolved phages. Why choosing evolution of 30 days? Did all the 30d-evolved phages show expanded host range?

Some other points may need further clarification:

1. Line 308-310: The sentence of “we co-cultivated two different lytic phages, phage Ace and phage APV, with several clinical strains of *Klebsiella pneumoniae* in 4 mL of Tris LB.” means in one tube (both phages) or two different tubes (one phage in one tube)? What are the exact clinical strains (these strains are susceptible to the wildtype phages or not) used? Again, are these strains together in one tube or separated in individual tubes?
2. Fig. S3: From the figure, it could see that the bacterial densities in 3 replicates were quite consistent during the days, while the phage densities in 3 replicates showed different patterns during the days. The trained phage also showed different dynamics to different isolates (Figure 4 and 5). Could these results mean that finding the phages with expanded host range is by chance since every replicate will generate different evolved phages?
3. Data discrepancies: a 1.69% variance exists between text-reported APV phage efficiency (28.81%, line 164) and graphical presentation in Fig. S2 (27.12%).
4. The scientific rationale for phage evolution requires strengthening: the selected parental phages (APV and Ace) showed moderate lytic activity (61.02% after 30-day evolution), while the pre-screened phage KL35 demonstrated superior lytic efficiency (72.88%, Figure 2S) without evolutionary manipulation. This raises questions about the necessity of the evolutionary approach.

Reviewer #2

(Remarks to the Author)

Reviewer #3

(Remarks to the Author)

Manuscript is quite well written and is of some interest. However the lack of any discussion of the capsular serotype or indeed the genotype of the *Klebsiella pneumoniae* isolates used is alarming. It is well known that tail fibre structures in *Klebsiella* phage have polysaccharide depolymerases and receptor binding sites to recognise and hydrolyse the capsule to allow access to the outer membrane of the host. Widening phage host range within a serotype is therefore constrained to a limited diversity of phage.

There are many reasons phages are not used in therapy currently, emerging resistance is not the main reason. It is largely a lack of safety data and regulatory approvals and in most species, narrow host range.

Reviewer #4

(Remarks to the Author)

The study proposed by Ghatbale and colleagues presents a successful protocol for training bacteriophages to enhance their host range against clinical *Klebsiella* strains isolated from UC San Diego Health. The manuscript is well written, clear, and notably concise.

However, I am uncertain about the originality of the work, as several previous studies have demonstrated similar findings, in which phage activity was enhanced following co-culture with bacterial hosts. The adaptation of an existing *Escherichia coli* protocol to *Klebsiella* may not, in itself, constitute a substantial scientific advance.

That said, the manuscript would benefit from a more detailed exploration of the biological mechanisms underpinning the adaptive evolutionary changes observed. In particular, the following points could be addressed:

What are the genetic characteristics of the clinical *Klebsiella* strains used in the study (e.g., sequence type, capsule type, O antigen)?

Does the phage training process broaden the host range sufficiently to enable infection of *Klebsiella* strains with distinct genetic backgrounds?

How stable are the mutations observed in phage genomes following adaptation?

Do these mutations confer increased environmental fitness to the trained phages?

A deeper investigation into these aspects would considerably strengthen the impact and scientific contribution of the study.

Version 2:

Reviewer comments:

Reviewer #1

(Remarks to the Author)

The revised manuscripts have addressed most of my previous concerns. The only thing left is still the novelty of the experimental phage evolution. Although the authors have explained that using clinical isolates is the main difference from the previous reports where the lab cultures were used, this difference is small. Furthermore, if using clinical isolates is important, no animal model experiments were done to demonstrate that the evolved phages could show better outcomes to treat the infections caused by the clinical isolates compared to the wild phages.

Reviewer #2

(Remarks to the Author)

Reviewer #3

(Remarks to the Author)

Manuscript has been re-written to, briefly, address my concerns. The paper is interesting and although not a major advance in the field it is not insignificant.

Reviewer #4

(Remarks to the Author)

The authors have revised the manuscript, taking into consideration the reviewers' suggestions. I only have two small

additional comments:

Please add the sequence type of the Kpn strains in Fig. S4.

Please update the phage taxonomy according to the new ICTV guidelines (Table S1 and throughout the manuscript). It is important that a publication of such high quality uses the most up-to-date taxonomy data available.

REVIEWER COMMENTS

Reviewer #1 (Remarks to the Author):

While the manuscript entitled “Experimental Phage Evolution Results in Expanded Host Ranges Against MDR and XDR *Klebsiella pneumoniae* Isolates” addresses the potentially important concept of expanding phage host range through evolutionary approaches to target clinically resistant bacteria, it unfortunately lacks sufficient novelty to warrant publication in its current form.

1. Except for using clinical isolates, the experimental methodology employed for phage evolution substantially overlaps with established protocols reported in previous studies (e.g. doi: 10.1126/science.adi5536). Actually, the hypothesis of broadening host range through phage co-evolution has been mentioned in previous studies, e.g. doi: 10.1126/science.adi5536 and doi: 10.1128/AEM.02138-20. It is not surprising to see that the evolved phages would expand host range.

We consider that the use of clinical isolates is a significant difference between our study and previous studies. Not only does it represent a departure from how most research is typically done, but it differs from how most research focuses on isolates that have been adapted to laboratory conditions for years prior to use in studies like these. Most of the isolates in this study we subjected to these conditions with minimal passages, which is a significant departure from established research protocols. We modified the manuscript to clarify that the co-evolution techniques were already established, but that our use of recent clinical isolates is a significant departure from most typical research studies. We addressed this in the text in lines 102 to 107:

“In this work, we focused on the pathogen *Klebsiella pneumoniae* and adapted existing co-evolutionary techniques to evolve phages that had implemented strategies to overcome the development of rapid *K. pneumoniae* resistance in clinical strains. In particular, we were seeking to evolve *Klebsiella* phages that had expanded and prolonged activity against MDR (multi-drug resistant) and XDR (extensively-drug resistant) clinical isolates of *K. pneumoniae*. “

We are aware that the co-evolution technique has been used previously, as mentioned by the reviewer. In the work presented by Borin J. et. al., 2023. (ref 17) the co-evolution was performed in lab adapted *Escherichia coli* and the model phage ϕ 21. In this work we challenged the field by using clinical strains and recently isolated phages. Dr. Justin Meyer, the corresponding author of the previous coevolution papers and a co-author of this manuscript.

We highlighted the differences between our study and previous studies; this is included in the manuscript in lines 150 to 154:

“The primary differences between the two protocols were the use of clinical isolates compared to laboratory-adapted isolates, the fact that *Klebsiella* isolates had mucoid

phenotypes compared to non-mucoid *E. coli*, and the evolution process between host and phage took place for a full 30 days before the phages were recovered”

We are also aware of the co-evolution approach used in *Listeria monocytogenes* by Peters T.L. et.al., 2020 and were already citing their work (ref 19) in lines 178-180:

“Such training experiments have previously been performed for *E. coli*, *K. pneumoniae*, *P. aeruginosa*, and *L. monocytogenes* [19-22].”

To our knowledge, this is the first time such a strategy has been undertaken with clinical strains rather than laboratory adapted strains, which is of high significance. It tells us that those concepts that we discover on the bench can be immediately translatable to both phages that are still circulating in the environment and clinical strains that are still circulating in our patients. It gets us so much closer to the bench-to-bedside concept that we are pursuing in our research.

2. The genetic mutations in the evolved phages were found related to the L-shaped tail fiber protein, affecting host recognition and binding mechanisms. The mechanism is also similar to that found in previous studies (e.g. doi: 10.1126/science.adi5536, doi: 10.1128/AEM.02138-20).

Yes, we found a similar mechanism and we are citing the above-mentioned papers, references 17 and 19.

3. Based on previous studies, co-evolution of phage and bacteria will generate a mixture of different evolved phages. Why choosing evolution of 30 days? Did all the 30d-evolved phages show expanded host range?

In previous studies from our group (Borin J. et al, 2023) it was shown that over a 21-day period a complex co-evolution network emerged, in which the phages were able to switch receptors. We decided to expand this period to 30 days to challenge this notion. Each co-evolution experiment was performed in triplicate, and in some replicates, we observed that after 21 days the phages were below the limit of detection of our PFU assay, however, we were able to recover phages at the end of the experiment (Figure S3). This shows that maintaining a co-evolution experiment for 30 days challenges both bacteria and phage adaptability.

Not all the co-evolved phages showed a host range expansion, and this is shown in Figure 3. The percentage of lysed strains in the ancestral phage APV was 28.8%. This host range increased in four experiments and decreased in two experiments. Similarly, the ancestral phage Ace lysed 42.3 % of the strains, which increased in three experiments, decreased in two experiments, and remained the same in one experiment.

The text was modified to incorporate this information explicitly, lines 167 to 177:

“For example, different trained phage APV isolates varied in their lytic capacity from having some lytic capacity against 16.95% of the isolates to 61.02% of the isolates,

overall, the host range of the co-evolved phages increased in four experiments and decreased in two experiments (**Fig. 3**). We found similar results for phage Ace, where the post-evolution phage isolates ranged in activity from 30.51% to 59.32% of the *K. pneumoniae* isolates, showing an expansion of host range in three experiments, a decrease in host range in two experiments, and remained the same in one experiment. There was host-range expansion identified against both ESBL isolates and non-ESBL isolates. These data strongly suggest that the 30-day process of phage-host evolution could result in a set of phages with improved capabilities in lysing their *K. pneumoniae* hosts”

Some other points may need further clarification:

1. Line 308-310: The sentence of “we co-cultivated two different lytic phages, phage Ace and phage APV, with several clinical strains of *Klebsiella pneumoniae* in 4 mL of Tris LB.” means in one tube (both phages) or two different tubes (one phage in one tube)? What are the exact clinical strains (these strains are susceptible to the wildtype phages or not) used? Again, are these strains together in one tube or separated in individual tubes?

The text was modified to explicitly state that they were individual experiments, and to include the list of strains used for co-evolution. Lines 311 to 314:

“Briefly, we co-cultivated each different lytic phage, phage Ace or phage APV, with each one of several clinical strains of *Klebsiella pneumoniae* (KP319, KP315, KP122, KP96, KP117, or KP112) in 4 mL of Tris LB”

All strains were susceptible to the phages at the beginning of the experiment. This was incorporated in the text, lines 315 to 316:

“All strains were susceptible to the phages at the beginning of the experiments.”

2. Fig. S3: From the figure, it could see that the bacterial densities in 3 replicates were quite consistent during the days, while the phage densities in 3 replicates showed different patterns during the days. The trained phage also showed different dynamics to different isolates (Figure 4 and 5). Could these results mean that finding the phages with expanded host range is by chance since every replicate will generate different evolved phages?

We did not test the host range or genomic changes on each one of the triplicates. The objective of doing triplicates was to have redundancy so that no individual replicate shaped the overall results and that extinction events may not completely shape overall trends. This was shown in Figure S3.

Each phage was trained in a different bacteria isolate. We did not find a correlation between the bacteria in which the phage evolved and host range expansion.

3. Data discrepancies: a 1.69% variance exists between text-reported APV phage efficiency (28.81%, line 164) and graphical presentation in Fig. S2 (27.12%).

Thanks for noticing this, the text was modified accordingly.

4. The scientific rationale for phage evolution requires strengthening: the selected parental phages (APV and Ace) showed moderate lytic activity (61.02% after 30-day evolution), while the pre-screened phage KL35 demonstrated superior lytic efficiency (72.88%, Figure 2S) without evolutionary manipulation. This raises questions about the necessity of the evolutionary approach.

There are several reasons we decided to perform the co-evolutionary protocol above and beyond just extending host ranges that were important for those who are in the phage therapy field. The first is that we were hoping greatly to improve the threshold for the development of resistance for these phages against the bacterial strains. This may be an underappreciated aspect of phage therapy, but often resistance develops rapidly, which limits the utility of phage therapy. If we can improve the threshold for resistance development through the co-evolutionary protocol, we can have a significant impact in the phage therapy field. We showed that we increased the threshold for resistance development by examining the longitudinal inhibition over 72 hours. These results are really impressive for host/phage interactions, and you don't typically see these types of experiments in phage/host papers because the authors know that resistance usually develops after 24 hours so they won't show what happens after 24 hours. We gladly show what happens up to 72 hours. This is very meaningful for what will probably happen in a human. The second is that we were hoping to expand host ranges. We chose not to focus on our best phages to start with because we believed that we had less room for growth in host ranges if we started with our best phages. That's why we started with APV and Ace because they had moderate lytic activities. Third, we had some concern about extinction events. We have previously performed these experiments in other species and when we start with phages that are too "good" at killing, we experience extinction events early. That reduces our opportunities for improvements in phage utility because we cannot observe co-evolution when extinction occurs quickly. We have briefly added these explanations to the text of the discussion of the manuscript. Lines 161-164:

“Host range expansion. We focused on two of our phages with relatively modest host ranges (**Fig. S2**), phages Ace and APV (belonging to the family Straboviridae) for our phage evolution/host range expansion experiments. We selected phages with modest host range to give them the opportunity to improve.”

Reviewer #2 (Remarks to the Author):

Thank you for your efforts in revising our manuscript, we appreciate it.

Reviewer #3 (Remarks to the Author):

Manuscript is quite well written and is of some interest. However the lack of any discussion of the capsular serotype or indeed the genotype of the *Klebsiella pneumoniae* isolates used is alarming. It is well known that tail fibre structures in *Klebsiella* phage have polysaccharide depolymerases and receptor binding sites to recognise and hydrolyse the capsule to allow access to the outer membrane of the host. Widening phage host range within a serotype is therefore constrained to a limited diversity of phage.

This is an important point the reviewer brings up. In response, we have modified the manuscript to include further discussion of the putative genotypic variation observed in the *Klebsiella* collection. Lines 177 to 183:

“Host range expansion was observed to *Klebsiella pneumoniae* clades for which the ancestral phages showed no lysis. Coevolved APV phages expanded their host ranges to four clades that were not lysed by the ancestral phage. Coevolved Ace phages expanded their host range to four clades that were not lysed by the ancestral phage (**Fig S4**). This host range expansion also included lysis of putative O-antigen and Capsule serotypes not previously infected.”

There are many reasons phages are not used in therapy currently, emerging resistance is not the main reason. It is largely a lack of safety data and regulatory approvals and in most species, narrow host range.

We apologize if we left the impression that emerging resistance was the only factor limiting phage therapy. We have clarified our prior remark to include safety, regulatory approvals, and narrow phage host ranges.

Lines 57-59:

“Treatments like bacteriophages have not had much success against such pathogens because of their narrow host range, and resistance to the phages used often develops rapidly.”

Lines 90-97:

“Phage technology has recently seen a resurgence in Western medicine [12] given the surge in antibiotic resistant microbes and the relative lack of available alternatives for physicians. Ongoing clinical trials evaluating the safety of phage therapy are an important step towards enabling its broader clinical application. Also, given recent advances in technology, scientists have been poised to make advances in phage technology that was not available when antibiotics were largely replacing them as the treatments of choice for bacterial infections.”

Reviewer #4 (Remarks to the Author):

The study proposed by Ghatbale and colleagues presents a successful protocol for training bacteriophages to enhance their host range against clinical *Klebsiella* strains isolated from UC San Diego Health. The manuscript is well written, clear, and notably concise. However, I am uncertain about the originality of the work, as several previous studies have demonstrated similar findings, in which phage activity was enhanced following co-culture with bacterial hosts. The adaptation of an existing *Escherichia coli* protocol to *Klebsiella* may not, in itself, constitute a substantial scientific advance. That said, the manuscript would benefit from a more detailed exploration of the biological mechanisms underpinning the adaptive evolutionary changes observed. In particular, the following points could be addressed:

1. What are the genetic characteristics of the clinical *Klebsiella* strains used in the study (e.g., sequence type, capsule type, O antigen)?

To address this question, we performed whole genome sequencing on all the clinical strains of this study. We performed phylogenetic analysis to determine sequence type, as well as O-antigen and capsule serotype predictions *in silico*. This was incorporated in the manuscript in lines 317 to 333:

“**Bacteria whole genome sequencing.** Genomic DNA was extracted from the clinical strains using DNeasy Blood & Tissue Kit (Qiagen, catalog number 69506). Whole genome sequencing was performed using the NovaSeq platform (Illumina), producing 150 bp paired-end read format (2 × 150 bp).

Bacteria genome analysis. Genomes were denovo assembled using SPades, annotations were performed using BV-BRC bacteria annotation recipe. Phylogenies were constructed using BV-BRC Bacterial Genome Tree tool which utilizes a codon tree method that selects single-copy BV-BRC PGFams and analyzes aligned proteins and coding DNA from single-copy genes using the program RAxML. Parameters were set to 1000 genes with 0 deletions or duplications. The phylogenetic clades and annotations were performed using iTOL. Isolates serotype classification was performed using the *in silico* program Kaptive, which takes genome assemblies and compares the K and O loci to a curated reference database to find the best matching locus. A scoring matrix is used that considers sequence identity, coverage, missing genes, and extra genes to determine the confidence level of the match between the query genome and the reference locus. “

As a summary of this analysis, Figure S4 was incorporated. Lines 653 to 661:

Figure S4. Genomic characterization of clinical strains of *Klebsiella pneumoniae*. Phylogenetic tree of the whole genomes using codon tree method. Bootstrap values are shown in black. Branch distances are shown in grey. Strains used in the coevolution experiments are shown with a blue circle. Eleven distinctive clades were determined. Capsule and o-antigen groups were determined using Kaptive. Infectivity matrix of ancestral and coevolved phages is shown. Regions where host-range expansion to clades not infected by the ancestral phage are shown in red dotted regions.

2. Does the phage training process broaden the host range sufficiently to enable infection of *Klebsiella* strains with distinct genetic backgrounds?

This was incorporated in the manuscript, lines 177 to 183:

“Host range expansion was observed to *Klebsiella pneumoniae* clades for which the ancestral phages showed no lysis. Coevolved phages APV expanded their host range to four clades that were not lysed by the ancestral phage. Coevolved phages Ace expanded their host range to four clades that were not lysed by the ancestral phage (**Fig S4**). This host range expansion also included lysis of O-antigen and Capsule serotypes not infected previously.”

3. How stable are the mutations observed in phage genomes following adaptation?

We were able to capture these mutations after a 30-day experiment. We continue to grow these trained phages in laboratory conditions, and we still see a similar infection patterns, which may imply that the mutations are still present.

4. Do these mutations confer increased environmental fitness to the trained phages?

These mutations were not found on the phages isolated from the environment; therefore, we think they are not the most ideal ones for environmental fitness.

A deeper investigation into these aspects would considerably strengthen the impact and scientific contribution of the study.

REVIEWERS' COMMENTS

Reviewer #1 (Remarks to the Author):

The revised manuscripts have addressed most of my previous concerns. The only thing left is still the novelty of the experimental phage evolution. Although the authors have explained that using clinical isolates is the main difference from the previous reports where the lab cultures were used, this difference is small. Furthermore, if using clinical isolates is important, no animal model experiments were done to demonstrate that the evolved phages could show better outcomes to treat the infections caused by the clinical isolates compared to the wild phages.

Unfortunately, our laboratory does not perform animal experiments because we do not feel they are a great approximation of the phage/host interplay that we observe in humans. We have, however, registered a clinical trial that is currently undergoing revisions with the FDA. It is already funded and is set to recruit 30 human subjects, with 6 of them having Klebsiella pneumoniae pulmonary infections. Those subjects will receive phage cocktails including multiple of the phages developed in this study. So, we will have some objective evidence of the efficacy of these phages in eradicating pulmonary infections in the years to come. Only those studies will be performed in humans rather than in animal models.

Reviewer #2 (Remarks to the Author):

We do not see any comments that require a response. Thank you for reviewing our manuscript.

Reviewer #3 (Remarks to the Author):

Manuscript has been re-written to, briefly, address my concerns. The paper is interesting and although not a major advance in the field it is not insignificant.

Thank you for reviewing our manuscript.

Reviewer #4 (Remarks to the Author):

The authors have revised the manuscript, taking into consideration the reviewers' suggestions. I only have two small additional comments:

Please add the sequence type of the Kpn strains in Fig. S4.

We have added these putative sequence types as suggested to Figure S4.

Please update the phage taxonomy according to the new ICTV guidelines (Table S1 and throughout the manuscript). It is important that a publication of such high quality uses the most up-to-date taxonomy data available.

We have updated the phage taxonomy according to the new ICTV guidelines as suggested in Table S1 and throughout as suggested. Thank you for pointing this out.